# Metabarcoding Analysis of Microorganisms Inside Household Washing Machines in Shanghai, China

**DOI:** 10.3390/microorganisms12010160

**Published:** 2024-01-13

**Authors:** Tong Chen, Shu Zhang, Juan Yang, Youran Li, Eiichi Kogure, Ye Zhu, Weiqi Xiong, Enhui Chen, Guiyang Shi

**Affiliations:** 1Key Laboratory of Industrial Biotechnology, Ministry of Education, School of Biotechnology, Jiangnan University, Wuxi 214122, China; 2KAO (China) Research and Development Center, No. 623, Ziri Road, Minhang District, Shanghai 100098, Chinazhu.ye@kao.sh.cn (Y.Z.); xiong.weiqi@kao.sh.cn (W.X.); chen.enhui@kao.sh.cn (E.C.); 3National Engineering Research Center for Cereal Fermentation and Food Biomanufacturing, Jiangnan University, Wuxi 214000, China; 4Jiangsu Provincial Research Center for Bioactive Product Processing Technology, Jiangnan University, Wuxi 214122, China; 5Kao Corporation, 1334, Minato, Wakayama 640-8580, Japan

**Keywords:** laundry contamination, microbial community, biofilm, metagenomics, detergent

## Abstract

Washing machines are one of the tools that bring great convenience to people’s daily lives. However, washing machines that have been used for a long time often develop issues such as odor and mold, which can pose health hazards to consumers. There exists a conspicuous gap in our understanding of the microorganisms that inhabit the inner workings of washing machines. In this study, samples were collected from 22 washing machines in Shanghai, China, including both water eluted from different parts of washing machines and biofilms. Quantitative qualitative analysis was performed using fluorescence PCR quantification, and microbial communities were characterized by high-throughput sequencing (HTS). This showed that the microbial communities in all samples were predominantly composed of bacteria. HTS results showed that in the eluted water samples, the bacteria mainly included *Pseudomonas*, *Enhydrobacter*, *Brevibacterium*, and *Acinetobacter*. Conversely, in the biofilm samples, *Enhydrobacter* and *Brevibacterium* were the predominant bacterial microorganisms. Correlation analysis results revealed that microbial colonies in washing machines were significantly correlated with years of use and the type of detergent used to clean the washing machine. As numerous pathogenic microorganisms can be observed in the results, effective preventive measures and future research are essential to mitigate these health problems and ensure the continued safe use of these household appliances.

## 1. Introduction

Washing machines are integral to modern life, playing a pivotal role in maintaining personal hygiene and cleanliness. However, the very appliances designed to eliminate dirt and microbes may harbor and propagate these microorganisms. Microbial contamination in washing machines has emerged as an area of concern due to its potential impact on human health [1]. In recent years, there has been a growing focus on the hygiene conditions of washing machines. A study conducted by Nix et al. in 2015 analyzed microorganisms in German washing machines and identified proteobacteria as the dominant bacterial microorganisms, whereas *Basidiomycetes* and *Ascomycetes* were the main colonizing fungi [2]. The China Center for Disease Control and Prevention conducted a study in 2016 that involved sampling 15 household washing machine water samples in Beijing to test for pathogenic bacteria and conditionally pathogenic bacteria such as *Pseudomonas aeruginosa*, *Staphylococcus aureus*, and *Escherichia coli*. Notably, the study found a detection rate of 66.67% for *Pseudomonas aeruginosa* [3]. Furthermore, in 2022, Li Jiaqing et al. investigated public washing machines in a college dormitory in Guiyang for four consecutive months and discovered microbial contamination and the presence of pathogenic bacteria [4].

Washing machines can become reservoirs for a diverse array of microbes. The general public is increasingly concerned about the issue of microbial contamination in washing machines. On the one hand, clothes and fabrics washed by washing machines are often in close contact with the body, and people are highly attentive to their cleanliness. On the other hand, washing machines have become an essential part of modern people’s daily lives. While washing machines provide convenience, the development of odors and mold after prolonged use can reduce the overall experience and sense of security. During the use of a washing machine, microorganisms can enter the machine through clothing, home textiles, and washing water [5]. Due to the enclosed structure of washing machines and the humidity in the usage environment, microorganisms tend to accumulate in areas such as the inner cylinder, the rubber seal, and the detergent dispenser, leading to cross-contamination during the washing process [6,7]. This issue poses a multifaceted challenge as it involves both the introduction of harmful microorganisms into the washing machine, typically from soiled clothing, and the subsequent dispersion of these contaminants onto previously clean garments. Such a process not only threatens the efficacy of the washing machine in achieving its primary objective but also raises significant concerns related to public health. In this context, it is crucial to delve into the various aspects of cross-contamination within washing machines [8].

Currently, there are two categories of methods for characterizing microbial communities in washing machines: traditional culture-dependent methods and culture-independent methods [9]. Each has its advantages and disadvantages. Usually, combined methods are used to obtain more comprehensive information on the structural composition and diversity of microbial communities [10]. The plate counting method is the basis for exploring the structural diversity of microbial communities through traditional culture-dependent methods. By using different media or culture conditions, the probability of microorganism separation can be effectively increased. To obtain a more comprehensive analysis of microbial diversity and achieve accurate identification of microbial communities (including those with low abundance), technologies are being widely applied in functional microbial ecology research. Compared with traditional sequencing techniques, high-throughput screening (HTS) techniques such as nano-biofilm arrays and droplet microfluidics have higher throughput and sequencing efficiency, save a lot of time, and can also provide more detailed, accurate, and reliable digital information, which helps to correctly understand the relationships between various microbial communities in the process of cross-contamination [11,12,13]. Combining multiple methods and collecting samples from different locations and times can provide a more comprehensive understanding of the microbial communities in washing machines. This information can be used to develop strategies to improve hygiene, reduce microbial contamination, and ensure the safe use of these appliances. Currently, there is a limited amount of research available on the microorganisms inside washing machines, and their community characteristics and potential harm to the human body are not well understood [14,15]. Based on the aforementioned background, this study aims to investigate the internal microbial community structure of washing machines, providing insights for the development of materials, products, and methods pertaining to washing machines.

## 2. Materials and Methods

### 2.1. Sample Collection and Preprocessing

This study was conducted from July to August 2022, when 22 private households with regular washing machine cleaning habits were selected for on-site screening and visits. The selection of washing machines was based on their conditions, usage patterns, cleaning methods, and the demographics of washing machine users. According to a 2016 survey by the Chinese Center for Disease Control and Prevention [1], bacterial contamination inside washing machines tends to increase with time, with no significant changes observed after three years. This suggests a potential influence of biofilm. Therefore, the duration of usage was classified into three groups: 1–3 years, 3–5 years, and over 5 years. Regarding washing machine usage, various habits can affect the hygiene of washing machines. This study categorized users based on factors such as the use of the drum self-cleaning mode, the addition of disinfectant during washing, the frequency of cleaning the washing machine, and the duration of soaking during cleaning. In terms of user demographics, the number of individuals using the washing machine might influence the rate of microbial accumulation. Therefore, users were divided into groups based on whether one generation, two generations, or three generations shared the same living space. As shown in Table 1, each sampled household was assigned a unique identity (ranging from 1001 to 1034 with intervals). Samples were collected from each household, including tap water, inner-barrel water, pure washing water, washing water with non-antibacterial detergent, and biofilm. When sampling the washing water with detergent, 30 g of Attack Instant Clean Liquid Detergent (KAO Corporation, Wakayama, Japan) was added into each washing machine, followed by washing according to the standard procedure, and then sampling. For water samples, 1000 mL of water was collected in a beaker disinfected with 75% ethanol and transferred to sterile sample bottles, which were then placed in an ice box for transportation. Upon arrival at the laboratory, the samples were concentrated through a 0.22 μm filter and stored at −20 °C until further analysis. Biofilm samples were collected from the outer wall, base, rubber ring, etc., of the inner cylinder using a cotton swab, which was then placed in an EP tube. The biofilm sample was stored at 0 °C until further testing.

### 2.2. Biomass Quantification through Either DNA Extraction or Plate Counting

For samples with high biomass content, the filter was shredded using 75% ethanol-wiped scissors and placed in a clean EP tube. DNA extraction was performed using a DNA extraction kit. After extraction, DNA concentration was measured using a NanoDrop spectrophotometer (NanoDrop Technologies, Wilmington, DE, USA), and the results were further tested through agar gel electrophoresis. The remaining extracted DNA fragments were stored in a −20 °C refrigerator for future use. For relatively high biomass samples, real-time fluorescence quantification based on molecular biology was employed [16]. For samples with low biomass content whose DNA could not be adequately extracted, microbial isolation counts were performed using relatively quantitative coating methods. To do that, one piece of filter was taken into conical flasks and mixed with 100 mL of sterile water. The mixture was then shaken for 30 min to obtain a suspension. Under aseptic conditions, 100 μL of the suspension was transferred onto solid potato dextrose agar (PDA) medium using a dilution spread method, ensuring even distribution across the solid medium. The Petri dishes were partially sealed with parafilm to facilitate aeration while still preventing rapid desiccation of the culture medium during the incubation process. They were inverted in a biochemical incubator (25 °C) until visible colonies appeared on the surface.

### 2.3. Biomass Determination Based on RT-qPCR

Real-time fluorescence quantitative polymerase chain reaction (RT-qPCR) is applied as a high-throughput microorganism detection technology. It involves the addition of fluorescent probes to the PCR reaction system, allowing for real-time monitoring of changes in fluorescence signals during the PCR reaction. This enables the detection and quantification of target products based on the establishment of a standard curve [17]. For quantifying bacteria, the 16S DNA of *Escherichia coli* DH5α was selected as the standard product. For fungal quantification, general fungal primers were used. The extracted plasmid DNA was ligated to a vector, and the concentration was measured using an ultra-micro-UV spectrophotometer. The number of plasmid DNA copies was calculated based on the relative molecular mass of the vector, and the corresponding numerical value was recorded.

### 2.4. High-Throughput Sequencing

Regions V4-V5 of the bacterial 16S rRNA gene were detected using the forward primer 515F (5′-GTGCCCGCMGGCGGGGGGGTAA-3′) and the reverse primer 907R (5′-CCGTCAATTCMTTTTTRAGTTT-3′) [18]. Generic primers were used to amplify fungal internal transcribed genes (ITS1F, 5′-CTGGTCTTAGAGAGAGAGGAGGAAGTAA-3′ and ITS2R, 5′-GTGCGTTCTCTCTCTCATCGGATGC-3′) [19]. Each 50 μL PCR volume contained 1 × PCR buffer solution (Mg^2+^), 0.2 mM dNTP, 0.4 mM forward primer, 0.4 mM reverse primer, and 1.25 U TaKaRa Taq HS polymerase (Dalian, China). A total of 10 ng of the sample genome was added to the above reaction PCR system. The PCR amplification procedure was pre-denaturation at 94 °C for 5 min, denaturation at 94 °C for 30 s, annealing at 55 °C for 30 s, and extension at 72 °C for 45 s. After a total of 32 cycles, it was maintained at 72 °C for 5 min. The genomic PCR products of each sample were analyzed through agarose gel (1.8% *w*/*v*) electrophoresis. Strips were removed from the agarose gel and the GelDnaPurification Kit (TaKaRa, Dalian, China) was used to purify and recover the DNA on each of them. After cutting and gluing, PCR products containing index sequences were purified using the SanPrep Column PCR Product Preparation Kit (Sangon Biotech, Shanghai, China). The purified PCR products were quantified with a Nanodrop@ ND-1000 UV-Vis ultraviolet spectrophotometer (Thermo Scientific NanoDrop, Waltham, MA, USA), then PCR products from different samples were mixed at equal molecular weights, and paired-end 2 × 300 bp sequencing was performed using the MiSeq platform and MiSeq kit v3 (Illumina, San Diego, CA, USA).

Genetic sequences obtained were processed using QIIME 2 [20]. Simply put, raw sequencing readings were assigned to specific samples using exact matches to barcode sequences and filtered to exclude low-quality sequences, namely those with <150 bp in length, mean Phred score <20, ambiguous bases, and/or single nucleotide repetition >8 bp. The remaining high-quality paired-end readings were assembled using FLASH [21]. After the detection and removal of chimera, the remaining high-quality sequences were clustered into amplified sequence variants. Classification was performed using the Q2 feature classifier QIIME 2 plug-in to implement the sklearn method and the pre-trained SILVA database (version 132) [22], with a similarity of 99%. All sequence data were deposited at the National Center for Biotechnology Information (NCBI) under accession numbers PRJNA1055691(16S) and PRJNA1055872(ITS). 

### 2.5. Confocal Microscopy Approaches to Analyze the Biofilm Matrix

Dimethyl sulfoxide (DMSO) was utilized to prepare storage solutions of 1 mg/mL and 2 mg/mL fluorescein isothiocyanate (FITC, Fujifilm, Wako, Japan) and Nile Red (APExBIO, Houston, TX, USA) separately. The hydrophobic filters were used to filter the dyes, and the solutions were stored in the dark at −4 °C. For FITC staining, 20 mg of biofilm sample was taken, and 1 mL of phosphate buffer solution (50 mM, pH = 9) was added, followed by the addition of 50 μL FITC. After dark incubation on a rotator for more than 8 h, the reaction was terminated by adding 5 M ammonium chloride. Subsequently, for Nile Red staining, 3 μL of 2 mg/mL Nile Red dye was added to the FITC-stained samples. After avoiding light exposure, staining was carried out for 5–10 min, and 10–20 μL was used for slide preparation. The samples were observed under excitation light at 488 nm and 552 nm. A confocal laser scanning microscope (CLSM, Leica TCS SP8, Wetzlar, Germany) was used to observe the biofilm activity, and 6–8 images were collected at random for each sample to avoid experimental error.

### 2.6. Statistical Analysis

Species diversity matrices are presented based on the binary Jaccard index. Principal coordinate analysis (PCoA) for beta diversity was constructed and visualized using the R package (R ade4 package, version 2.15.3). The significance between groups in the PCoA plot was tested with permutational multivariate analysis of variance (PERMANOVA). Multiple comparisons and heat mapping were also performed using the R package. Linear discriminant analysis (LDA) effect size (LEfSe) analysis was used to determine the biomarkers explaining the differences between the samples from different washing machines. LDA score cut-off was set at 3 [23]. FAPROTAX and FUNGuild were used to infer the functional profiles of the microbiota communities. All data were standardized during the statistical analysis.

## 3. Results

### 3.1. Microbial Content of Washing Machine Water Samples

Using fluorescence quantitative PCR and microbial pure culture methods, we detected the total number of bacteria and fungi in tap water to be 60 CFU/100 cm^2^ and <2 CFU/100 cm^2^, respectively. In the pure water elution group, the total number of bacteria was 310 CFU/100 cm^2^, with 29 CFU/100 cm^2^ for fungi. The washing solution elution group showed higher counts, with the total number of bacteria exceeding 700 CFU/100 cm^2^ and fungi exceeding 50 CFU/100 cm^2^. The microbial colonies in all samples were predominantly composed of bacteria. The detergent was able to remove more microorganisms.

### 3.2. Analysis of Microbiological Composition of Household Washing Machines

Water samples from the washing machines were annotated using the Greengenes database for bacterial 16S rRNA gene identification. The samples exhibited a classification into 44 families, 121 lineages, 324 orders, 590 families, and 1420 genera. For fungal annotation, the ITS gene was classified using the UNITE database [24]. The overall classification of the samples included 51 groups, 108 classes, 226 orders, 418 families, and 628 genera. By classifying the household washing machine samples, it was observed that the bacterial operational taxonomic units (OTUs) detected in each household ranged from 971 to 3636, with 19 common OTUs across all households. Regarding fungal OTUs, they ranged from 19 to 399, with 5 common OTUs found in all households. In all sampled household washing machines, bacterial diversity was significantly higher than fungal diversity. Furthermore, distinct microbial community characteristics were observed among samples from different washing machines.

Previous studies proved that the specific locations of the washing machine are critical factors influencing the distribution of microbial communities [25,26]. Therefore, all water samples were grouped according to their sources, including TAPW—tap water, ROLW—water inside the drum, EW—empty washing water, DW—water before detergent cleaning, and DWA—water after detergent cleaning. The top 10 most abundant bacteria and fungi in the washing machine water samples were classified separately (Figure 1). At the genus level, the predominant fungal genera were *unclassified Fungi* and *Cyphellophora*. As illustrated in Figure 1a, *unclassified Fungi* dominated abundances in the TAPW, ROLW, and EW groups, constituting 56.47%, 29.37%, and 17.12%, respectively. In the DWA group, *Cyphellophora* exhibited the highest abundance at 48.98%, with the second-highest in the DW group at 18.72%. In contrast, *Exophiala* exhibited the highest abundance exclusively in the EW group at 21.06%. Compared to the freely circulating fungal microbial communities in the water samples, substantial differences were observed in the fungal composition of biofilms. Among the top ten abundances identified in all 19 successfully detected samples, only four genera overlapped with major fungal genera in water samples: *Aspergillus*, *Rhodotorula*, *Exophiala*, and *Cyphellophora* (Figure 1b). Additionally, *Fusicolla* and *Fusarium* were prominent in the biofilm-associated fungal community. Regarding bacterial distribution, the top ten most abundant genera were listed for both the water and biofilm samples (Figure 1b,c). *Pseudomonas* stood out as the most abundant bacterial genus in TAPW and DW, consistently comprising over 10% in all the samples. *Enhydrobacter*, another major bacterial genus, was prevalent in ROLW, EW, and DWA. Intriguingly, its presence was notably low in TAPW (0.49%), suggesting a source distinct from the washing machine’s incoming water. Comparing bacterial diversity between the water and biofilm samples, seven genera were consistently present in the top ten abundances in both groups, differing from the fungal scenario. Furthermore, the bacterial community in the biofilm samples exhibited a concentrated distribution, with more than 20% abundance in each sample. Notably, samples M1033 and M1001 showcased this concentration, with *Enhydrobacter* and *Brevibacterium* reaching 77.18% and 80.05%, respectively. Additionally, the top 10 bacterial microbial genera in the biofilms accounted for 53.8% of the total community, which was ten times higher than the corresponding index for free microorganisms (5.2%). The overall annotation results of species composition showed similarities in the microbial species composition of washing and dehydration, regardless of whether detergent was used or not. However, potentially pathogenic microorganisms such as *Pseudomonas* and *Acinetobacter* were found [27,28]. The findings underscore the distinct microbial dynamics in washing machine biofilms, emphasizing the need for further investigation into potential sources and implications for public health. 

### 3.3. Analysis of Differences in Various Water Samples

Principal coordinate analysis was performed to analyze the differences between tap water and washing water with detergent. As shown in Figure 2a, the results showed significant differences in bacteria families, indicating that tap water is not the primary source of microorganisms in washing machines. Another principal coordinate analysis was conducted comparing water samples inside the drum (ROLW) and empty washing water (EW) (Figure 2b). The findings revealed no significant differences between bacteria and fungi in the water samples before and after washing, indicating that microorganisms in the washing machine strongly adhere to surfaces. Pure water, when combined with mechanical forces alone, cannot effectively remove these microorganisms. Considering the similarity in bacterial diversity between the samples inside the drum and in pure washing water, the two were combined as a pure water elution group (NT). Principal coordinate analysis was performed to compare this group with the detergent elution group (XD). As shown in Figure 2c,d, the results demonstrated clear differences in bacteria when detergent was present versus when it was not, while no significant differences were observed in fungi. This indicates that bacteria easily remain within the washing machine even after washing. 

### 3.4. Correlation Analysis of Microbial Communities in Washing Machines

Several factors, including the years of use of the sampled washing machines, the use of the bucket self-cleaning mode, the addition of disinfectant during daily washing, the frequency of using laundry tub cleaner, the dosage form of the cleaner, the washing frequency, and the household composition, were considered. The microbial community in the washing machines was classified and analyzed based on these factors to identify the factors influencing microbial composition.

The years of use of the washing machines were categorized as 1–3 years, 3–5 years, and 5 years or more (Figure 3a). Principal coordinate analysis revealed that the microbial community composition evolved with the years of use (Figure 3b,c). Samples older than 5 years displayed the lowest microbial diversity according to the sparseness curve. This could be attributed to the effects of biofilm adhesion, where microorganisms are not easily eluted after prolonged use. It is also possible that dominant strains expand more prominently during the evolution process, occupying more living space and suppressing other strains. In microbial communities, certain strains can outcompete others, leading to their dominance. Dominant strains may have specific adaptations that allow them to thrive in a given environment [29]. Over time, if these dominant strains continue to reproduce and occupy more ecological niches, they can suppress the growth of other, less competitive strains. This phenomenon can further reduce overall diversity within the community [30,31,32,33].

Significant differences were observed between bacterial communities when using effervescent tablets and powders as washing machine cleaners (Figure 3d,e). Powdery cleaners exhibited better cleaning effects, possibly due to the larger surface area of the powder, enabling fuller contact with the surface of the washing machine inner cylinder and improving cleaning efficacy [34]. Contrary to expectations, even among households that habitually add disinfectant during the washing process, no significant difference was found in the microbial community composition of the washing machines. This indicates that the use of disinfectants does not effectively address microbial contamination in washing machines. 

### 3.5. Function Prediction of the Microbial Communities in Biofilm Samples

In our investigation of microbial communities within the biofilm samples of washing machines, we conducted FAPROTAX functional prediction analysis on bacterial communities. This exploration aims to understand the potential functions of microbes colonizing the internal environment of washing machines and their potential impact on users. FAPROTAX is a prokaryotic functional annotation database compiled manually from the literature on cultivable bacteria. It encompasses over 7600 functional annotations across more than 80 functional groups, such as nitrate respiration, methane production, fermentation, and human pathogens, collected from over 4600 prokaryotic microorganisms. As seen in Table 2, a number of microorganisms are involved in crucial biogeochemical processes and interspecies interactions. The putative functions mainly include the biogeochemical cycles of microorganisms, especially the circulatory functions of sulfur, carbon, hydrogen, and nitrogen. Among them, the most common functions are chemoheterotrophy and aerobic chemoheterotrophy. These two functions are mainly contributed by abundant bacteria such as *Brevibacterium* and *Acinetobacter*. It is noteworthy that human pathogens consistently rank among the top ten predicted major functions of bacteria in all biofilm samples, averaging over 5%. This underscores a significant potential health risk associated with the biofilm inside washing machines. These pathogenic bacteria can be transmitted to clothes during the washing process. Subsequent contact with these contaminated clothes may lead to skin infections or other health issues. Individuals with weakened immune systems, such as the elderly or those with pre-existing health conditions, are particularly vulnerable to infections caused by pathogenic microorganisms from biofilms. After dual staining with FITC and Nile Red, the biofilm samples were prepared, and images were captured using confocal laser scanning microscopy under different conditions. As depicted in Figure 4, variations in the distribution of proteins and lipids were observed in the biofilm samples. Sample 1001 exhibited bright, clustered green fluorescence, suggesting the presence of a higher concentration of viable bacteria in these regions. Conversely, most biofilm samples displayed weak green fluorescence, indicating a limited number of viable bacteria. Under excitation at a wavelength of 552 nm, these samples often exhibited distinct red fluorescence, which may be attributed to the breakdown of lipid components from accumulated, long-deceased bacterial bodies.

### 3.6. Correlation Analysis of Microbial Communities

We compared the predominant microbial communities in the biofilm and water samples. The results revealed that the top 10 most abundant microorganisms in the biofilm constituted over 70% of the total microbial population, a proportion significantly higher than that observed in free-living microorganisms in the water samples (Figure 5a,b). Furthermore, major microbial genera present in the biofilm, such as *Enhydrobacter*, *Acinetobacter*, *Pseudoxanthomonas*, and *Brevibacterium*, were also detected among the dominant genera of free-living microorganisms in the water samples, suggesting potential microbial migration between water and biofilm. These findings suggest that not all microorganisms can extensively colonize the biofilm; only a subset of free-living microorganisms in the washing machine actively participates in biofilm formation. Charles also reported that certain microorganisms have characteristics or mechanisms that facilitate their adherence and colonization within the biofilm environment [35], while others may not possess these traits. This selectivity could be influenced by factors such as microbial species-specific attributes, surface properties, and the biofilm’s microenvironment. The distribution of biofilm microorganisms exhibited distinct temporal patterns. In cases where the washing machine’s service life exceeded five years, the biofilm was highly dominated by *Enhydrobacter* and *Acinetobacter* species (Figure 5c). Conversely, in biofilm samples from shorter durations (1–3 years), these two bacterial genera did not exhibit clear dominance. These temporal patterns suggest that the composition of the biofilm microbiome in washing machines changes over time. This is an important insight because it implies that the microbial communities adapt and evolve in response to the conditions within the machine. This finding also raises questions about the mechanisms that drive the dominance of *Enhydrobacter* and *Acinetobacter* species in older washing machines. Further research is needed to investigate whether these species have specific adaptations that make them more competitive in this environment or if other factors are at play. 

## 4. Discussion

The widespread use of shared laundry machines globally presents potential health hazards, particularly regarding microbial migration. Commonly located in residential and communal spaces, these facilities may inadvertently contribute to the dissemination of harmful microbes, impacting public health. The water and surfaces within the machine create an environment conducive to microbial exchange and transfer, potentially leading to the persistence of pathogenic bacteria and the dissemination of antibiotic resistance genes. Tap water and washing machines were sampled from 22 households in Shanghai, and the microorganisms in the washing machine environment were analyzed. Although it is reported that one of the most significant concerns with tap water is the presence of harmful microorganisms, such as bacteria, viruses, and parasites [36,37], quantitative analysis revealed a significantly higher number of microorganisms in the water samples from washing machines compared to tap water. This indicates that tap water is not the main source of microbial contamination in washing machines. The metagenomic analysis based on the Illumina platform identified a total of 3550 bacterial types and 911 fungal types in the samples. Notably, potential pathogens such as *Pseudomonas* spp. and *Acinetobacter* spp. were also detected. Thus, washing machines harbor a high number and diverse range of microorganisms, including many pathogenic species. *Pseudomonas* spp. and *Acinetobacter* spp. are known to cause skin infections. *Pseudomonas aeruginosa*, for instance, can lead to conditions like folliculitis and hot tub rash when it comes into contact with the skin [38]. In a washing machine, these pathogens may transfer onto clothing, potentially leading to skin irritations and infections upon wear. Inhalation of aerosolized droplets containing these pathogens, especially when garments contaminated with *Pseudomonas* spp. or *Acinetobacter* spp. are worn, may lead to respiratory issues. Individuals with pre-existing respiratory conditions, such as asthma or chronic obstructive pulmonary disease (COPD), could be particularly vulnerable to exacerbations of their conditions [39]. People with compromised immune systems, such as those undergoing chemotherapy or organ transplant recipients, are at heightened risk. These pathogens, if present in washing machines, could pose a more severe threat to individuals with weakened immune responses. Many factors, including the concentration of microbes, individual susceptibility, and the presence of other pathogenic species, influence the actual risk of illness. However, these potential health implications underscore the importance of maintaining clean washing machines, adopting proper laundry hygiene practices, and conducting further research to better understand and mitigate the risks associated with microbial contamination in these appliances.

By considering factors such as the years of use, the self-cleaning mode, the disinfectant usage, the washing frequency, the dosage form of the cleaner, and the household composition, it was observed that microorganisms in washing machines evolve with changes in usage time. Longer-term use and complex household structures were associated with increased health hazards. The decline in microbial diversity over time may be due to biofilm formation, making it more difficult for microorganisms to be eluted. Biofilms are complex communities of microorganisms encased in a self-produced matrix of extracellular polymeric substances (EPSs). In the context of washing machines, biofilm formation can exacerbate the presence and persistence of harmful microorganisms, such as bacteria and fungi, with several key implications. First, it acts as a reservoir for pathogenic microorganisms within washing machines. These biofilm communities provide a protected environment where microbes can thrive, shielded from detergents and disinfectants used during regular washing cycles. Additionally, dominant strains may expand, limiting the space available for other strains. Moreover, households with a more diverse composition exhibited higher microbial diversity in their washing machines, suggesting a link between population structure and microbial communities. Microbes within biofilms are continuously shed into the wash water during each laundry cycle. This shedding can introduce contaminants into clean laundry, promoting cross-contamination and the spread of pathogens. To mitigate the influence of biofilm formation on microbial contamination in washing machines, regular cleaning and maintenance are crucial. Proper cleaning routines can help disrupt and remove biofilms, reducing the risk of cross-contamination and microbial persistence. Additionally, manufacturers can explore materials and design modifications that are less conducive to biofilm development, ultimately contributing to cleaner and safer washing machine environments.

The investigation into microbial migration within washing machines draws attention to a potential pathway for the transmission of antibiotic resistance genes (ARGs). Microbial communities thriving in biofilms, a common occurrence in washing machines, act as reservoirs for ARGs. The migration of these microbes during washing cycles raises concerns about the dissemination of antibiotic resistance. Firstly, biofilms may house bacteria carrying ARGs, which encode resistance to antibiotics frequently used in households. Secondly, microbial migration through the water during washing cycles allows potentially antibiotic-resistant organisms to contact various fabrics. The water serves as a medium for transporting both microbes and any genetic material they carry. In addition to migration, the potential for horizontal gene transfer among microbes in the washing machine environment is a pivotal factor. This mechanism facilitates the exchange of genetic material, including ARGs, between different microbial species. Subsequently, the contamination of clothes and other laundered items may contribute to the spread of antibiotic resistance beyond the confines of the washing machine environment. This poses a potential risk to public health, as individuals may inadvertently come into contact with and transport antibiotic-resistant microbes and genes. Consequently, it is imperative to develop strategies aimed at minimizing the spread of antibiotic resistance in household settings and mitigating its potential impact on public health. Understanding the dynamics of microbial migration and gene transfer in washing machines is essential for devising effective cleaning practices and designing machines that reduce microbial retention, thereby safeguarding public health.

Despite the presence of disinfectants, there was no significant difference in microbial diversity in washing machines between households that habitually added disinfectant and those that did not. This indicates that disinfectants alone are insufficient to effectively address microbial contamination in washing machines. The comparison of bacterial and fungal diversity between inner-barrel water and pure washing water demonstrated that the mechanical force of washing without detergent has little effect on removing microorganisms from the washing machine. This suggests that microorganisms in the washing machine have strong adhesion. Furthermore, bacteria tend to concentrate in washing machines, and neither pure water nor detergent can thoroughly eliminate microorganisms from the machine. Disinfectants may struggle to penetrate and eradicate microbial populations within biofilms. Effective disinfection often requires sufficient contact time between the disinfectant and the target microorganisms. In washing machines, the short duration of a typical wash cycle may not provide adequate time for disinfectants to exert their full antimicrobial effect, especially on microorganisms sheltered within biofilms or textiles. On the other hand, the use of strong disinfectants in washing machines can have environmental consequences. Residual disinfectants in wastewater can impact aquatic ecosystems and contribute to antimicrobial resistance in the environment.

In conclusion, washing machines harbor microbial contamination that poses a potential risk to human health. Bacteria easily remain in the washing machine even after washing. The types of microorganisms in the biofilm are more concentrated in a few taxa, and this phenomenon becomes more pronounced with the passage of time. The currently available washing products on the market are not effective at thoroughly cleaning the washing machine. Future research should focus on utilizing the distribution and abundance of microorganisms, particularly pathogenic bacteria, in washing machines to develop materials and technologies aimed at controlling the hygiene conditions inside the machines. 

## Figures and Tables

**Figure 1 microorganisms-12-00160-f001:**
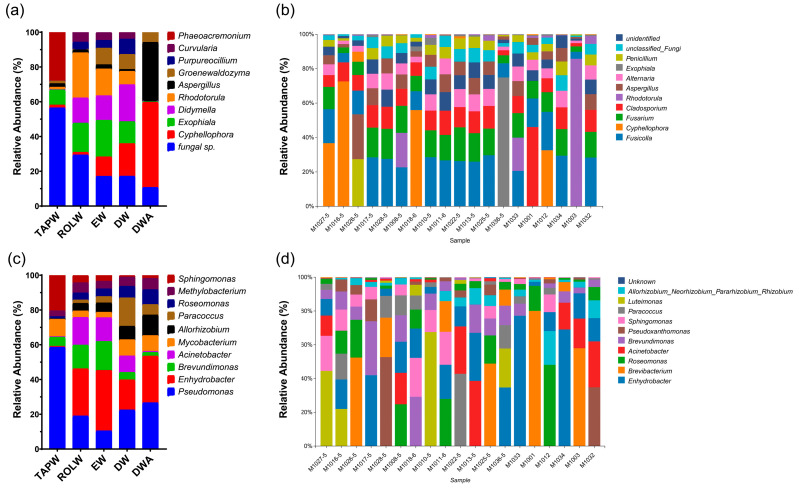
The microbial community structures of samples collected from household washing machines. (**a**) Stacked bar chart showing fungal genus composition of water samples based on relative abundance data, grouped according to their source parts; (**b**) fungal genus composition of biofilm samples based on relative abundance data; (**c**) bacterial genus composition of water samples based on relative abundance data, grouped according to their source parts; (**d**) bacterial genus composition of biofilm samples based on relative abundance data.

**Figure 2 microorganisms-12-00160-f002:**
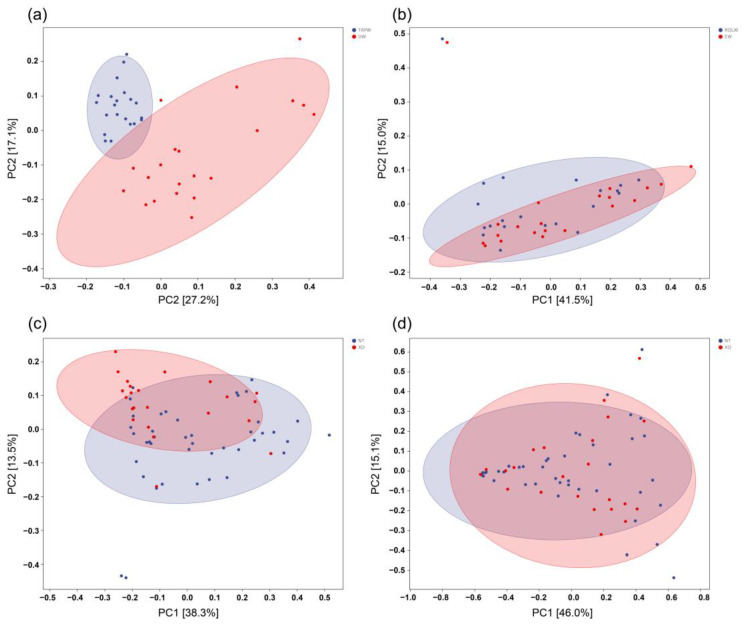
A score plot of the principal component analysis of bacterial and fungal taxa identified in different samples. Eclipses in the principal component analysis were made at the 95% confidence interval using singular value decomposition (SVD) computation. (**a**) Bacterial diversity between tap water (TAPW) and washing water with detergent (DW); (**b**) bacterial diversity between water samples inside the drum (ROLW) and empty washing water (EW); (**c**) bacterial diversity between pure water elution group (NT) and detergent elution group (XD); (**d**) fungal diversity between pure water elution group (NT) and detergent elution group (XD).

**Figure 3 microorganisms-12-00160-f003:**
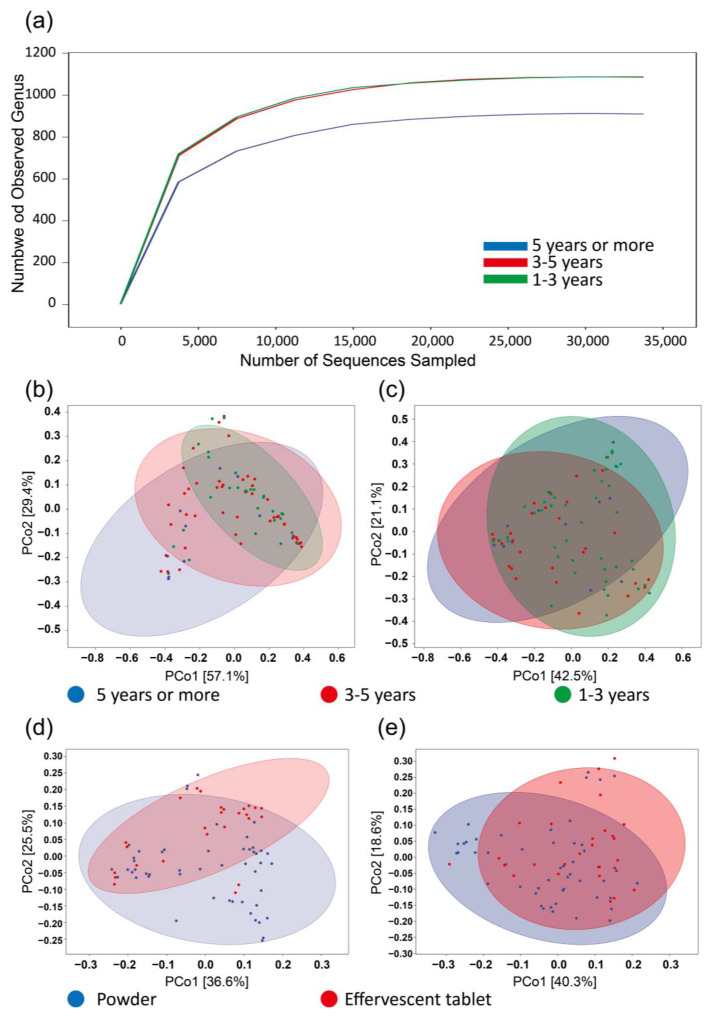
Washing machine microbiomes clustered by years of use and washing machine cleaner types. (**a**) Rarefaction curves show the number of unique operational taxonomic units (sharing ≥97% sequence identity) per total reads for each sample; (**b**) bacterial diversity across different years of use of the washing machines; (**c**) fungal diversity across different years of use of the washing machines; (**d**) bacterial diversity between powder group and effervescent tablet group; (**e**) fungal diversity between powder group and effervescent tablet group.

**Figure 4 microorganisms-12-00160-f004:**
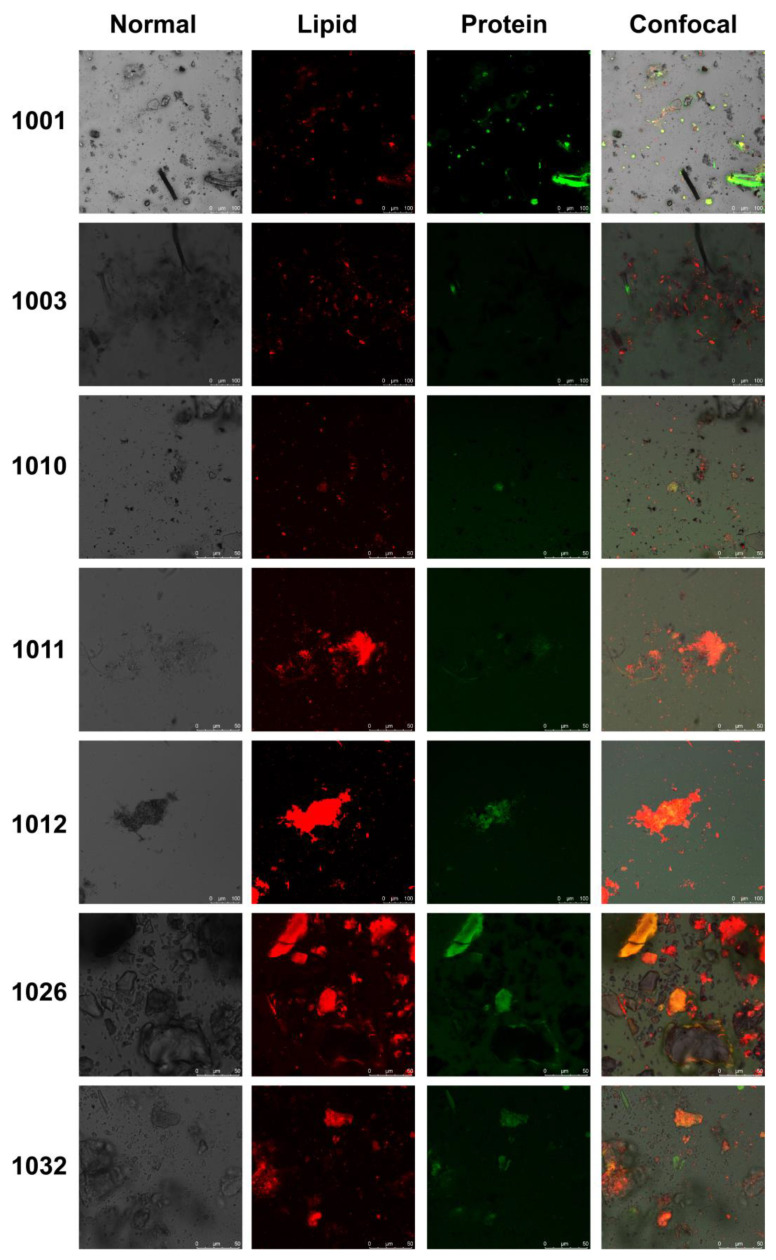
CLSM datasets of biofilm samples from washing machines. Lipids were stained with Nile Red, and proteins were stained with fluorescein isothiocyanate (FITC). The sample identities are indicated on the left side of each set of images.

**Figure 5 microorganisms-12-00160-f005:**
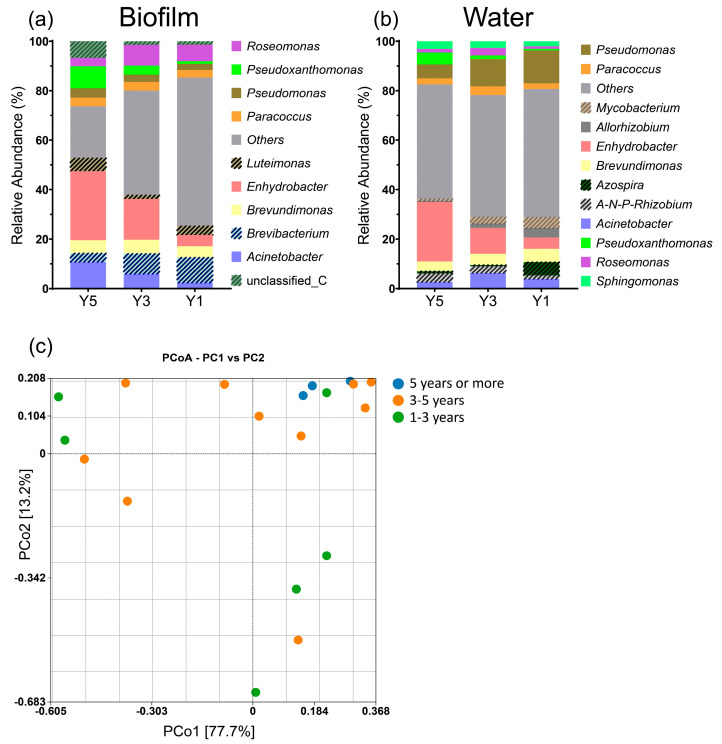
Biofilm and water samples’ microbiomes clustered by years of use. (**a**) Stacked bar chart showing bacterial genus composition of biofilm samples based on relative abundance data, grouped according to years of use, unclassified_C, *unclassified_Comamonadaceae*; (**b**) stacked bar chart showing bacterial genus composition of water samples based on relative abundance data, grouped according to years of use, *A-N-P-Rhizobium*, *Allorhizobium-Neorhizobium-Pararhizobium-Rhizobium*. The 7 high-abundance bacteria shared between the biofilm and water samples are represented in the same pure color in both groups. Bacteria exclusively in either the biofilm or water samples are color-coded with diagonal fill patterns. (**c**) Bacterial community composition by principal coordinate analysis (PCoA). Blue, orange, and green circles indicate 5 years or more, 3–5 years, and 1–3 years of use, respectively.

**Table 1 microorganisms-12-00160-t001:** Information about the collected samples ^1^.

Household Identity	Family Members	Type of WM ^2^	Service Life (Years)	Usage Frequency (Per Week)	WM Cleaning Frequency	Type of WM Drum Cleaner	Detergent Soaking Time
1001	Couple + 1 child	Drum	1–3	3–5	1–3	Powder	>2 H
1003	Couple + parents + 1 child	Drum	3–5	6–7	1–3	Liquid	<30 min
1005	Couple	Drum	3–5	3–5	1–3	Powder	>2 H
1008	Couple + parents + 1 child	Agitator	1–3	6–7	<3	Powder	<30 min
1010	Couple + 1 child	Drum	1–3	6–7	1–3	Powder	<30 min
1011	Couple + 1 child	Drum	3–5	6–7	1–3	Powder	>2 H
1012	Couple + 1 child	Drum	3–5	3–5	1–3	Effervescent tablet	<30 min
1013	Couple + parents + 1 child	Drum	3–5	6–7	1–3	Effervescent tablet	<30 min
1016	Couple + 1 child	Agitator	1–3	3–5	<3	Powder	<30 min
1017	Couple + 1 child	Drum	3–5	6–7	<1	Powder	<30 min
1018	Couple	Agitator	1–3	3–5	<3	Powder	<30 min
1022	Couple + 1 child	Drum	3–5	3–5	<1	Effervescent tablet	<30 min
1023	Couple + parents + 1 child	Drum	1–3	2–3	1–3	Effervescent tablet	>2 H
1025	Couple + 1 child	Drum	1–3	3–5	1–3	Effervescent tablet	<30 min
1026	Couple + 1 child	Drum	3–5	6–7	1–3	Powder	<30 min
1027	Couple + parents + 1 child	Drum	3–5	6–7	1–3	Effervescent tablet	>2 H
1028	Couple	Drum	3–5	3–5	1–3	Effervescent tablet	<30 min
1031	Couple + 1 child	Drum	5	6–7	<6	Powder	<30 min
1032	Couple + 1 child	Drum	5	6–7	3–6	Effervescent tablet	<30 min
1033	Couple	Drum	3–5	2–3	1–3	Powder	<30 min
1034	Couple	Drum	5	6–7	3–6	Powder	<30 min
1036	Couple	Drum	5	3–5	<6	Powder	<30 min

^1^ Standard washing was conducted by adding water or detergent, and four samples were collected from each washing machine, including tap water, water from the inner barrel, pure washing water, and washing water with non-antibacterial detergent. Additionally, by disassembling the washing machines, one biofilm sample was collected from each machine. ^2^ WM, washing machine.

**Table 2 microorganisms-12-00160-t002:** Putative functions of the microbial communities in biofilm samples.

	M1001	M1003	M1012	M1032	M1033	M1034	M1008-5	M1010-5	M1011-6	M1013-5	M1016-5	M1017-5	M1018-6	M1022-5	M1025-5	M1026-5	M1027-5	M1028-5	M1036-5
chemoheterotrophy	38.57	34.27	21.13	25.20	30.19	29.22	21.77	34.74	29.77	23.77	25.41	30.76	34.94	16.79	29.06	32.25	32.70	25.98	21.75
aerobic_chemoheterotrophy	37.80	28.60	19.77	25.01	30.04	29.09	19.13	32.46	26.46	23.58	23.90	30.19	31.87	16.34	28.66	24.31	30.79	25.39	20.12
fermentation	1.02	3.83	2.86	4.78	21.12	17.13	4.32	0.94	5.99	6.85	3.42	17.72	3.77	1.76	0.18	0.30	2.81	1.37	8.78
ureolysis	5.02	0.38	13.17	2.66	1.42	0.42	7.56	1.62	11.72	1.13	4.80	1.15	4.33	0.47	3.34	11.75	6.60	1.06	2.15
animal_parasites_or_symbionts	4.48	5.13	11.99	12.12	1.04	5.99	7.74	3.73	8.46	12.37	3.24	3.30	2.19	3.76	4.38	4.40	4.42	0.94	1.33
human_pathogens_all	4.47	5.12	11.97	12.12	1.04	5.99	7.65	3.72	8.46	12.37	3.24	3.30	2.08	3.57	4.38	4.40	4.42	0.93	1.33
nitrate_reduction	1.16	1.45	1.08	1.00	1.27	2.40	2.73	0.73	0.08	2.27	2.96	3.14	1.58	5.03	10.03	1.62	0.90	4.60	3.75
aromatic_compound_degradation	0.09	5.51	0.10	9.21	0.28	3.90	3.32	1.36	0.04	9.36	0.44	0.61	1.14	3.96	0.57	0.72	4.31	3.22	1.33
methylotrophy	0.81	0.04	2.27	0.31	1.26	0.04	3.73	0.59	3.36	0.24	4.26	0.51	3.76	4.70	0.72	7.44	2.12	3.49	4.47
methanol_oxidation	0.81	0.04	2.27	0.31	1.26	0.04	3.73	0.59	3.36	0.24	4.26	0.51	3.76	4.70	0.72	7.44	2.12	3.49	4.47
Other	5.75	15.64	13.38	7.28	11.08	5.78	18.31	19.53	2.31	7.82	24.06	8.82	10.58	38.92	17.95	5.38	8.79	29.53	30.52

## Data Availability

Data are contained within the article.

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
