# Peer review of "Metabarcoding Analysis of Microorganisms Inside Household Washing Machines in Shanghai, China"

_microorganisms, 2024, doi:10.3390/microorganisms12010160_

Round 1
Reviewer 1 Report
Comments and Suggestions for Authors
I accept the manuscript after major revision. The research work is interesting and applicable for human. The research devoted to explore the origin of the micro-organism either from tap water or washing machine and a recent research was focus on 22 household in china. However, some point must taken in consideration before publication
1-The detection of micro-organism through florescence technique can be illustrated in the revised manuscript with more information about the mechanism
2-The authors indicated that biofilms resist the production of micro-organism, more information require on the types and mechanism of bio-films in removal of these toxic organisms
3-Are the authors investigate the effect of different types of bio-films on the destruction of these micro-organisms
4-The authors can explain the effect of different types of washing products on the existence of these harmful micro-organisms
5-Are the exposure of the washing clothes to natural sunlight with 7% UV radiations has relevant effect on destructing these micro-organism
6-What is the best mechanism for your point of view on the perfect mechanism for destructing organic pollutant
7-More recent references are required to add to the revised manuscript
Comments on the Quality of English LanguageThe English language is quite good require only a careful revision of final version of the manuscript
Author Response
Regarding the response to the peer review comments, please refer to the attached document.

Reviewer 2 Report
Comments and Suggestions for Authors
The work of Tong Chen and co-workers characterizes the microorganisms found inside household washing machines from Shanghai, China.
I suggest that the authors consider the following points in preparing the final manuscript.
Minor comments:
1. Lines 70 – 71. The method name "non-culture-in-dependent methods" is incorrect. It should be "culture independent".
2. Line 73. „The plate coating culture method” - Did the authors mean “The plate counting method’?
3. Line 223. „fungal sp.” – it is not a name of genera.
Major comments:
1. First of all, the title is incorrect. In my opinion it should be “Characteristics of microorganisms ………” or “Analysis of microorganisms …..”.
2. Lines 21 – 22. The authors claim that “It showed that the microbial communities in all samples were composed of bacteria, and they have a strong adhesion ability in the washing machine environment.” Although they did not examine the adhesive properties of microorganisms.
3. Lines 99 – 102. The authors categorized the washing machines into two groups: drum washing machines and agitator washing machines. What was the purpose of this categorization if no analysis was carried out in this respect?
4. Lines 116 – 117. „ When sampling the washing water with detergent, the same type of detergent without antibacterial properties was used to control variables.” Please explain how this control was carried out.
5. Lines 120 – 122, line 130. “For water samples, 1000 mL of water was collected in a beaker disinfected with 75% ethanol and transferred to sterile sample bottles, which were then placed in an ice box for transportation.” In my opinion, the authors should have used a sterile beaker and sterile scissors.
6. Lines 135 – 144. Please let me know if these samples were also frozen after filtration.Why the Petri dishes were sealed with parafilm? This may have hindered the growth of obligately aerobic bacteria.
7. Lines 195 – 200. How did the authors convert the number of bacteria from a volume of 1000 mL to 100 cm2? Are the data given for one trial or the average of all trials?Lines 201 – 203. „While the detergent was able to remove more microorganisms, there was not a significant difference in magnitude, indicating the strong adhesion of microorganisms in household washing machines.” On what basis were these conclusions formulated??
8. Lines 115 – 116, 219 – 221. No correlation in sample description.
9. Descriptions of tables and figures should be detailed. The reader should not guess what data is presented. Descriptions should be adequate to the presented data, e.g.: lines 373 – 374 „High-abundance bacteria present exclusively in one group are depicted 373 with shaded regions of different colors.” Shaded regions are not visible in the figure.
10. The discussion and conclusions drawn on the basis of the research conducted are too far-reaching. The authors collected very diverse samples for their research (Table 1) and did not take into account all these differences in the analysis. In my opinion, the samples were too diverse. Moreover, there is no explanation of how the authors managed to compare the results obtained using different methods (classical microbiology and genetic methods). Was it necessary to use both types of methods?
Author Response

(The authors gave the same response as above.)

Reviewer 3 Report
Comments and Suggestions for Authors
Comments:
The submitted paper “Analysis of characteristic microorganisms inside household washing machines from Shanghai, China” by Chen et al. is interesting, comprehensive and focuses on an important thematic, which is in-house contamination and human health.
The paper is well written but could be better organised. Please see below my comments. The article needs some improvements to be worthy of publication in my opinion. But I liked the study!
Firstly, I would change the title to: “Metabarcoding analysis of microorganisms inside household washing machines from Shanghai, China”
The Abstract is clear, concise, pointing out the main results.
Keywords: I would change some of them, as some are listed in the title, so do not repeat them in these both sections (title and keywords).
The Introduction is well written, and the objectives are clearly stated; the introduction is a fair state of the art and points out why this study is still needed and important.
Line 42. Replace “colonized fungi” by “colonizing fungi”.
Methods are appropriate and are more or less well explained. The used techniques and methodologies were adequate for the study. But the statistical analyses as they are described do not reflect in a deep manner many analyses that were done to analyse the data and to present the results. Therefore, I would complete the section “2.5. Statistical analysis”. Correlations and PCAs need to be more detailed in this section, in my opinion.
My first question is: why 22 washing machines? In table 1, I see more than 22 samples (actually, 24 samples, being sample 1011 repeated (two different soaking times), and also sample 1018 with the same conditions is repeated. Why? This must be explained better! The authors analysed washing machines but from each one several samples were taken! I would complete the table 1. Also, the “groups” of samples that authors mention in the text of section 2.1 are not indicated in Table 1, as far I can see.
Another issue is the water before entering the machines. The tap waters were analysed and this is very clear to me. But I would prefer the complete description of the samples that were really analysed in the Methods section in Table 1. Moreover, are all the tap waters the same? And are these in the same conditions? I would recommend to have the characterization of the tap waters in terms also of chemical and physical parameters.
Results are in general well and extensively described with the appropriate number of tables and figures and supplementary materials, but the figures/images need to be enlarged to become more comprehensive.
Section 3.2 of the Results: Authors claim that they identified 3,550 species of bacteria and 911 species of fungi. This is absolutely wrong. Base on HTS the identifications cannot reach the species level, only genus level. I do know that there are many published HTS works with identification to species level but this is not accurate! We cannot insist in the same error! Please, keep the analysis only till the genus level!
Some parts of the Results are Discussion’s related. For example, lines 217 and 218.
Also, in lines 300-304…and 313-316. The same in lines 356-363
Authors can consider to fuse the Results and Discussion section. For me, it would be useful, as some issues are immediately discussed as the results are presented and described.
Line 224. Change “fungal sp.” By “fungi”.
Line 233. Change “bacterial species” by “bacteria”.
Another point: the sequences have to be deposited in public databases. Nothing is said about this.
Moreover, there are some points in the Results that were not addressed in the Materials and Methods section. For example, section of the Results 3.5. This must be completed.
Discussion of the results is extensive but some points presented in the Results should be transferred to here. The first part of the Discussion is somewhat a repetition of the introduction. Please revise. And again, I do think that in this work the authors could fuse the sections of results and discussion into only one section (Results and Discussion).
The conclusions are too much brief and could be more developed in my opinion.
The list of references is satisfactory and it supports well the paper, but it could be enlarged in my opinion.
Comments on the Quality of English Language
English needs minor editing.
Author Response

(The authors gave the same response as above.)

Round 2
Reviewer 3 Report
Comments and Suggestions for Authors
Dear Authors
Thank you for the corrections made. In my opinion, the article was much improved. Nevertheless, the phrase "The sequences were deposited in NCBI databases with a No. of SUB14105322" is absent in the paper. The nucleotide sequences accession numbers have to be available to the readers! This information is still missing in the paper and can be placed in the "Materials and Methods", at the end of section 2.4.
There is an extra dot in line 240, after "Fungi". Please correct.
Kind regards.
Author Response
Dear Expert,
Thank you very much for your encouraging and positive evaluation of our work. We were very pleased to note that you appreciated the revision and we thank you for their constructive review.
The sequencing information has been supplemented in the manuscript:
"All sequence data were deposited at the National Center for Biotechnology Information (NCBI) under the accession number PRJNA1055691(16S) and PRJNA1055872(ITS)."
Also, the extra dot in line 240, after "Fungi" has been removed.
Best wishes!